# Epigenetic Reader Bromodomain-Containing Protein 4 in Aging-Related Vascular Pathologies and Diseases: Molecular Basis, Functional Relevance, and Clinical Potential

**DOI:** 10.3390/biom13071135

**Published:** 2023-07-15

**Authors:** Xiaoxu Zheng, Kotryna Diktonaite, Hongyu Qiu

**Affiliations:** 1Center for Molecular and Translational Medicine, Institute of Biomedical Science, Georgia State University, Atlanta, GA 30303, USA; zhengxiaoxu.pku@gmail.com (X.Z.); kdiktonaite1@student.gsu.edu (K.D.); 2Department of Internal Medicine, Translational Cardiovascular Research Center, College of Medicine-Phoenix, University of Arizona, Phoenix, AZ 85004, USA

**Keywords:** aging, bromodomain-containing protein 4, epigenetic, vascular function, diseases

## Abstract

Aging is a key independent risk factor of various vascular diseases, for which the regulatory mechanisms remain largely unknown. Bromodomain-containing protein 4 (BRD4) is a member of the Bromodomain and Extra-Terminal domain (BET) family and is an epigenetic reader playing diverse roles in regulating transcriptional elongation, chromatin remodeling, DNA damage response, and alternative splicing in various cells and tissues. While BRD4 was initially recognized for its involvement in cancer progression, recent studies have revealed that the aberrant expression and impaired function of BRD4 were highly associated with aging-related vascular pathology, affecting multiple key biological processes in the vascular cells and tissues, providing new insights into the understanding of vascular pathophysiology and pathogenesis of vascular diseases. This review summarizes the recent advances in BRD4 biological function, and the progression of the studies related to BRD4 in aging-associated vascular pathologies and diseases, including atherosclerosis, aortic aneurism vascular neointima formation, pulmonary hypertension, and essential hypertension, providing updated information to advance our understanding of the epigenetic mechanisms in vascular diseases during aging and paving the way for future research and therapeutic approaches.

## 1. Introduction

Aging is a primary risk factor for vascular pathologies and diseases, such as atherosclerosis, aortic aneurysm, hypertension, aortic stiffing, and vascular neointima formation, and various mechanisms are involved in aging-related vascular diseases [1,2,3,4,5,6,7,8,9,10]. Despite distinctive phenotypes and different clinical manifestations, increasing evidence reveals that many changes associated with aging coexist in various vascular diseases [11], indicating a common regulatory mechanism underlying the aging vascular pathological procedure, leading to new research that is interesting to investigate the fundamental mechanisms involved in aging-related vascular diseases and therapeutic targets for aging-induced pathologies.

Bromodomain-containing protein 4 (BRD4) is a member of the Bromodomain and Extra-Terminal domain (BET) family, including bromodomain-containing protein 2 (BRD2), bromodomain-containing protein 3 (BRD3), and bromodomain testis-specific protein (BRDT) [12,13,14,15,16,17]. Similar to other BET family members, BRD4 contains two bromodomains (BDs) that can bind to acetylated histones [17]. It also has an extended C-terminal domain with little sequence homology distinguishing it from other BET members [17] that has been implicated in promoting gene transcription through interaction with transcription elongation factors [18,19,20]. It has been reported that BRD4 is a novel histone acetyltransferase (HAT) that acetylates histones H3 and H4 with a pattern distinct from other HATs. BRD4 acetylates histone 3 lysine 122 acetylation (H3K122), a residue critical for nucleosome stability, resulting in nucleosome eviction and chromatin de-compaction. Nucleosome clearance by BRD4 occurs genome-wide, including at its targets *myelocytomatosis oncogene (MYC)*, *FBJ murine osteosarcoma viral oncogene homolog (FOS)*, and *AURKB* (*Aurora B* kinase), resulting in increased transcription [21]. It has been found that BRD4 is ubiquitously expressed in various cells and tissues, participating in transcriptional elongation, chromatin remodeling, DNA damage response, and alternative splicing by recruiting different cofactors, regulating multiple cellular functions [15].

Studies indicate that BRD4 is required to regulate cell differentiation and tissue formation. Depletion of BRD4 causes abnormal mitosis with a high incidence of lagging chromosomes. Genes that control cell cycle progression and differentiation are found to be controlled by BRD4. For instance, BRD4 is necessary to maintain the identity of mouse and human embryonic stem cells (ESCs). Short hairpin RNA of BRD4 or treatment with BET inhibitors can cause ESCs to accumulate in the G1 phase of the cell cycle and take on the appearance of developing cells. BRD4 was also found to be able to control cell identity genes. For example, it is shown that BRD4 controls the expression of the octamer-binding transcription factor 4 (*Oct4*) gene, which is crucial for ESC identity; it is also attracted by the Oct4 protein to certain regulatory sites in the ESCs, such as those governed by the long non-coding RNAs that control X chromosome inactivation [22]. BRD4 was also found to be necessary for adipogenesis and myogenesis in vivo using tissue-specific BRD4 knockout (KO) mice [23,24]. BRD4 regulates the induction of cell-specific genes during adipogenesis by binding preferentially to active enhancers and BRD4 deletion results in a significant decrease in both BAT and muscle mass [23,24]. Myogenesis and the expression of myocyte genes are decreased by BRD4 knockdown in C2C12 myoblasts [23,24].

Clinically, BRD4 acts as a chromatin-binding protein implicated in cancer and autoimmune/infectious diseases that functions as a scaffold for transcription factors at promoters and super-enhancers. BRD4 has been found to be upregulated in many cancers to modulate oncogenic activities and cancer cell malignancy [25,26,27,28]. Thus, targeting BRD4 has been well-studied as a potential therapeutic method for cancers. Additionally, recent preclinical studies have identified the indispensable role of BRD4 in mouse hearts and highlighted the therapeutic benefits of BET BD inhibition in various types of heart disease, including myocardial infarction (MI) or transverse aortic constriction (TAC)-induced heart failure (HF), gene mutation-induced dilated cardiomyopathy (DCM), and dyskeratosis congenital (DC) [29,30,31].

Despite the significant findings in other diseases, much fewer studies of BRD4 were conducted in vascular pathology and the relative knowledge hold behind. Until recently, some studies found that BRD4 is abnormally expressed in many vascular cells and tissues and is highly associated with the pathogenesis of various aging-related vascular diseases [32]. In this review, we updated the recent advances in BRD4 biology that may be related to the development of vascular aging pathologies and summarize the progress in studies of BRD4 in aging-associated vascular diseases. This collected information will bring new insights into the understanding of the epigenetic regulatory mechanism in vascular diseases and the development of new therapeutic strategies in the future.

## 2. Molecular Basis for the Biological Functions of BRD4

Although the biological functions of BRD4 are cell- and tissue-dependent and there are still a lot of the regulatory mechanisms that have not been fully known, the progress in this area has provided valuable new information to invigorate future studies. Here, we highlighted the relative studies related to the BRD4 basic molecular structure and biological function as an epigenetic reader as well as the functional-associated modifications in BRD4.

### 2.1. Molecular Structure

As shown in Figure 1, human BRD4 structurally consists of multiple domains including two bromodomains (BD1 and BD2), an extra-terminal (ET) domain, and a histone acetyltransferase (HAT) catalytic domain. Within these domains, it contains a few specific functional sites including a nuclear localization signal (NLS) site, an N terminal cluster of phosphorylation sites (NPS), a C terminal cluster of phosphorylation sites (CPS), as well as the unique motif/residues, such as a C-terminal motif (CTM), two conserved motifs (A and B), an interaction domain with a high concentration of basic residues, and a Ser/Glu /Asp-rich region (SEED) with a concentration of serine residues mixed with glutamic and aspartic acid residues [22] (Figure 1).

Each BD has four alpha helices to form a hydrophobic pocket that enables anchoring the protein to an acetylated lysine residue, which allows BRD4 to specifically bind to acetylated lysine residues on histone H2, H3, and H4 with high affinities [33,34,35]. Between 600 amino acids (aa) and 678 aa, there are three helices in the ET domain, which is what distinguishes the BET family. Although its function has not yet been fully described, evidence suggests that the ET domain of BRD4 is a location for protein interaction with the BRD4 partners, including the histone demethylase, jumonji domain containing 6 (JMJD6), and a member of the nuclear remodeling and deacetylase repressor complex, chromodomain-helicase-DNA-binding protein 4 (CHD4), regulating transcription. The BRD4 ET domain could recruit proteins, including nuclear receptor binding SET domain protein 3 (NSD3), oxygenase JMJD6, and glioma tumor suppressor candidate region gene 1 (GLTSCR1), to form a complex that enables a positive transcription elongation factor (P-TEFb)-independent transcriptional activation [36,37]. The ET domain was also shown to interact with nipped-B-like protein (NIPBL), regulating transcriptions of many genes related to a rare growth-delay disorder, Cornelia de Lange Syndrome [38]. In addition to a range of cellular proteins, the ET domain guides viral protein γ-retroviral integration to the transcription start sites and enhancers through bimodal interaction with chromatin and γ-retroviral integrase (IN) [39].

BRD4 is considered a histone acetyltransferase also because it has a HAT catalytic domain at the C-terminal end. The CTM function has not been well studied, it was reported to form protein–protein interactions in the complex. BRD4 contains two intrinsic enzymatic functions in addition to acting as a scaffold for several transcription factors. It has been shown that BRD4 is an unusual kinase whose activity is mapped to its N terminal domain; deletion of the regions from the N terminal to the ET domain eliminates the kinase activity. The heptad repeat in the RNA Pol II central terminal domain is one of the BRD4 substrates, and BRD4 phosphorylates it specifically on Ser2 residues. Unexpectedly, BRD4 also contains intrinsic HAT activity as shown in the BRD4 structures. The consensus acetyl-CoA binding site for human BRD4 is found between 175 and 180 aa while the catalytic domain is situated distal to the ET domain in the region between 1122 and 1161 aa. While the homologous acetyl-CoA binding site and catalytic domain are present in both human and mouse BRD4, the mouse BRD4 possesses an extra acetyl-CoA (AcCoA) binding site in the C terminus at (1097–1102 aa) [22]. BRD4 HAT activity is eliminated by point mutations at either location. As a result, locations similar to AcCoA binding sites are where BRD4′s HAT activity is mapped. Mammals secrete the AcCoA binding protein (ACBP), impacting overall lipid metabolism and autophagy. Human body mass index (BMI) and ACBP levels are inversely correlated, demonstrating a direct relationship to the extra AcCoA binding site, and have the potential of contributing to a more gradual effect on mice compared to humans.

NPS and CPS are two important phosphorylation targets that could alter the function of BRD4. Phosphorylation of BRD4 by protein kinase casein kinase 2 (CK2) releases the BD2 domain from self-binding to NPS and is active in targeting the chromatin acetylated histone sites. While dephosphorylation of BRD4 by protein phosphatase 2A (PP2A) changes the conformation of BRD4 by masking BD2 with NPS, which leads to an inactive state with impaired target recognition and binding [40]. According to a report, a photo- switch mechanism controls BRD4 binding to chromatin, and the recruitment of p35 to transcription sites, through the basic residue-enriched interaction domain (524–580 aa) and NPS. BRD4 is also thought to interact with nucleosomal DNA through the two conserved A and B conserved motifs. Additionally, a study found protein kinase casein kinase 2 (CK2) phosphorylates the SEED domain, causing BRD4 to modify its conformation, which in turn affects its function. Lastly, the human BRD4 CTM region (1325–1362 aa) extends over the regions and interacts with P-TEFb [22].

### 2.2. Biological Function of BRD4 as an Epigenomic Reader

Although studies have shown that BRD4 plays distinct roles in various cells and tissues, the increasing evidence indicates that the most significant biological role of BRD4 in most cells is recognized as an epigenomic reader participating in the histone acetylation, gene transcription, and alternative splicing, which we summarized below.

#### 2.2.1. Histone Acetylation

BRD4 is thought to assist chromatin de-compaction through histone acetyltransferase activity and association with other histone acetyltransferases, deacetylases, and chromatin remodelers. BRD4 binds to acetylated histones at enhancers, promoters, and transcriptional start sites and aids in the formation of super-enhancer (SE) sites through its roles in chromatin decompaction, neutralizing positive charges on histones and other acetylated lysines, and recruiting the transcriptional machinery needed for rapid and abundant transcription [41]. Evidence shows that BRD4 maintains its role in recruiting transcriptional machinery to these latent enhancers, perpetuating aberrant gene expression in response to disease stimuli [41]. Further studies have revealed that not only does BRD4 play an important role in gene expression and gene regulation through the binding of promoters and nucleate super-enhancers, but BRD4 also vigorously regulates the process of transcription elongation [21]. A recent discovery illustrates BRD4 has intrinsic HAT activity, which plays a crucial role in controlling chromatin structure and has helped to clarify the function of BRD4 in maintaining chromatin structure and histone acetylation. Except for histone 3 lysine 14 acetylation (H3K14), which is acetylated by all other known histone 3 acetyltransferases (H3 HATs), BRD4 HAT activity is unique from that of other HATs. Importantly, BRD4 also acetylates histone H3 at the K122 residue, which is situated at the location on the nucleosome’s dyad axis where DNA-histone interactions are most powerful. Around the promoters of genes including *Myc*, *Fos*, and *AURKB* which is known to target and regulate, BRD4 selectively remodels chromatin and decreases nucleosome occupancy. Additionally, enhanced transcription is accompanied by global remodeling surrounding promoters caused by BRD4 overexpression [39].

#### 2.2.2. Gene Transcription

Transcription factor (TF) binding sites are comprised within an enhancer DNA element, in return creating a bridge for the accession of transcriptional regulatory complexes. BRD4 is present at enhancers which could correlate the linkage to TFs being affiliated with its function within the sites [42]. In one instance, the recruitment of BRD4 to enhancers or promoters could be influenced by histone acetylation patterns, which are chartered by the recruitment indirectly of TF-mediated acetyltransferase. However, BRD4 and TF interaction has a direct influence on recruiting BRD4 to chromatin. Recent studies have shown that BRD4 and certain TFs engage in direct contact with one another, as a result, develop in a BD-dependent or -independent matter. In addition to being present in tens of thousands of typical enhancer elements, BRD4 also functions as the nucleator of super-enhancers, which are collections of enhancer elements that are particularly enriched in transcription factors and coactivators like mediators and BRD4 and frequently span large kilobases. The majority of super-enhancers are oncogene-enriched or restricted to a certain cell type, together with the corresponding master transcription factors. BRD4’s cell-type specificity in the control of transcription is probably in large part a result of its interaction with super-enhancers. For instance, super-enhancers inside the MYC locus are connected to BRD4 in several malignancies, including acute myeloid leukemia (AML) and multiple myeloma [22].

BRD4′s presence at enhancers suggests a connection to the associated TFs. TFs recruit coactivators, leading to the acetylation of enhancer-associated nucleosomes and TFs themselves. BRD4 recruitment to enhancers may depend on histone acetylation patterns established by TF-medicated acetyltransferase recruitment or direct interactions with TFs. Recent studies show that BRD4 engages with specific TFs in a bromodomain-dependent or -independent manner. These interactions and the regulatory effects of BRD4/TF interactions may contribute to the therapeutic transcriptional effects of BET inhibitors in certain diseases. [42]. Differentiating between BD1 and BD2 pharmacologically offers a potential strategy to achieve more targeting transcriptional effects. A recent investigation has identified BET Inhibitors that exhibit a preference for BD2 and BD1. These compounds have been found to induce minimal transcriptional effects compared to JQ-1, indicating their potential for selective modulation of gene expression. Recently, a biochemical screen was conducted to determine if BRD4 interacts directly with several other purified factors, such as pre-initiation complex components, chromatin regulators, and sequence-specific TFs [40]. This investigation demonstrated that BRD4 directly binds to a particular subset of TFs in addition to verifying the interaction with P-TEFb that is known to occur. Since the TFs were produced and purified from E. coli, it is likely that these interactions took place in an acetylation-independent manner. The *Myc/Max* heterodimer, p53, ying yang 1 (*YY1*), *c-Jun*, adipocyte protein 2 (*AP2*), and CCATT-enhancer-binding protein (*C/EBP)* were among the BRD4-interacting TFs in this cohort [40]. BRD4 failed to interact with many of the TFs in the studies, including the potent activator Gal4-VP16, indicating that BRD4 has an inherent binding preference for specific TFs. Directly stimulating RNA-Pol II carboxy-terminal domain (CTD) phosphorylation at serine 2 to promote effective transcription is the function of BRD4. Additionally, cyclin-dependent kinase 9 (CDK9) (of the P-TEFb complex—a key regulator of transcription elongation and involved in controlling the expression of many genes across different biological processes) has been demonstrated to be phosphorylated by BRD4, which controls its enzymatic activity. BRD4 has been identified as a HAT that acetylates histones H3 and H4 in a manner that differs from that of other HATs. Increased transcription is caused by chromatin deconstruction and nucleosome eviction caused by BRD4 acetylating H3 K122 [40].

#### 2.2.3. Alternative Splicing

Recent research reveals that BRD4 controls RNA splicing through both its ET domain and HAT activity. To encourage the transcription of splice variants, BRD4 interacts with alternative exons and the RNA splicing regulator heterogeneous nuclear ribonucleoprotein M (HnRNPM) [31]. BRD4 travels with the elongation complex during transcription. A study investigated whether BRD4 is involved in the control of alternative splicing because most alternative splicing events occur concurrently with transcription. The conditional deletion of BRD4 during thymocyte development in vivo is linked to specific patterns of alternative splicing [43]. Similar to this, BRD4 knockdown in T cell acute lymphoblastic leukemia (TALL) cells change splicing patterns. Exon skipping is the most common alternative splicing event that BRD4 affects. Importantly, BRD4 co-localizes on chromatin with the splicing regulator, fused in sarcoma (FUS), and interacts with parts of the splicing machinery as determined by proximity ligation assays (PLAs) and immunoprecipitation (IP). The study suggests that BRD4 interacts with the splicing apparatus during transcription elongation, resulting in patterns of alternative splicing [43]. In TALL cancer cells and during thymocyte development in vivo, certain patterns of alternative splicing are linked to BRD4 depletion [43].

### 2.3. Functional Relevance of the Modifications in BRD4 Molecule

In addition to the basic molecular structure, BRD4 is also found to be undergone molecular modifications, which further alters its function in the cells and induced various effects in physiological and pathological conditions. Several post-translational modifications (PTMS) have been identified in BRD4 showing functional relevance. Recent studies also revealed that BRD4 has at least two alternative splicing isoforms with distinctive functions.

#### 2.3.1. Post-Translational Modifications (PTMs) of BRD4

Currently, ubiquitination and phosphorylation are found to be the main PTMs of BRD4; the former primarily controls the stability of the BRD4 protein and mediates BD and ET domain inhibitor (BETi) resistance, while the latter is connected to the biological functions of BRD4 such as transcriptional regulation, cofactor recruitment, and chromatin binding. BRD4 ubiquitination controlled by the speckle-type BTB/POZ protein—de-ubiquitination enzyme 3 (SPOP-DUB3) switch has been linked to aberrant degradation of BRD4 leading to BET-BD inhibitor resistance [15]. Interestingly, SPOP mutants in different types of cancer yield opposing effects on BET protein degradation by differential ubiquitination and sensitivity to BET-BD inhibitors [15].

NPS and CPS are the two phosphorylation domains of BRD4 that regulate BRD4 activities [44]. Hyperphosphorylation of BRD4 is often associated with an increased activity that could be found in the breast, colorectal, small-cell lung, pancreatic, and many other cancers [45,46,47]. Enhanced BRD4 activity may be due to the dimerization of two BRD4 monomers via binding negatively charged phosphorylated NPS to positively charged bromo-interacting domain (BID) regions with each other [44]. The conformational change would bring the BD2 and ET domains to chromatin to facilitate transcription modulations. Phosphorylation of BRD4 promotes dimerization and inhibits interaction with DNA and the formation of liquid–liquid phase separations (LLPS) while being necessary for active gene transcription. It may be possible that phosphorylated and unphosphorylated BRD4 create alternative molecular connections in LLPS due to the seemingly contradictory actions of phosphorylation in boosting gene transcription while decreasing LLPS in the chromatin [48].

In addition, the acetylation of BRD4 enhances its interactions with other proteins. For example, P300/CBP-associated factor (PCAF) acetylates transcription factor intestine-specific homeobox (ISX) at *Lys69* and BRD4 at *Lys332*, which forms a complex and translocates into the nucleus to promote transcriptional expression of genes important for epithelial–mesenchymal transition and metastasis [45]. Other PTMs such as hydroxylation and methylation all contribute in different ways to the control of BRD4. By controlling the expression of genes related to tumors, the variety, complexity, and reversibility of posttranslational modifications influence the structure, stability, and biological function of the BRD4 protein, as well as play a role in the occurrence and development of tumors as their primary and undeniable mechanism. Consequently, it may be useful to target BRD4-related alteration sites or enzymes to prevent cancer and give treatment [49].

#### 2.3.2. Alternative Splicing of BRD4

Human BRD4 is known to exist in two primary isoforms, namely the bromodomain protein 4 long isoform (BRD4L) and the bromodomain protein 4 short isoform (BRD4S). These isoforms are generated through splicing from a single gene located on chromosome 19. BRD4L encompasses a full-length structure with additional protein regions, including the CTM. In contrast, BRD4S lacks the CTM and HAT domains but retains functional domains such as the phosphor-dependent interaction domain (PDID) and basic interaction domain (BID). These differences in structure give rise to distinct functional roles and interactions for BRD4S compared to BRD4L, influencing gene expression and cellular processes [15].

The long isoform, BRD4L has been extensively studied in various cancers and other diseases, including transcription factors recruitment, transcriptional elongation, super-enhancer assembly, DNA damage response [15], and alternative splicing [43]. On the other hand, the short isoform, BRD4S was considered a less functional isoform because of the absence of the HAT domain. However, recent studies have shown that it alleviates breast cancer mortality and promotes human immunodeficiency virus 1 (HIV-1) latency [26]. Additionally, studies on LLPS have demonstrated the critical role of BRD4S in forming nuclear condensation in chromatin, regulating the gene transcription [48]. Variable cell types have different ratios of the long and short isoforms of BRD4 [11]. The mechanism of regulating alternative splicing of BRD4 has not been well studied, but serine/arginine-Rich Splicing Factor protein 1 (*SRPK1*) inhibition affects the global RNA splicing and BRD4 isoform levels [50], which implies that *SRPK1* could be involved in the BRD4 transcripts regulation process.

Research is currently underway to investigate the functional distinctions between BRD4S and BRD4L. In human cancer cells, BRD4S forms nuclear puncta with liquid-like properties and co-localizes with BRD4L, mediator complex subunit 1 (MED1), and histone H3K27ac. The presence of BRD4 puncta correlates with the expression of BRD4S but not BRD4L. Importantly, BRD4S plays a more significant role than BRD4L in incorporating BRD4 condensations into the chromatin [15]. This process is facilitated by the tandem BDs of BRD4S, which bind to lysine-acetylated histones. Furthermore, the function of BRD4S has been elucidated in the context of human papillomaviruses (HPVs). BRD4S lacks the CTM but retains domains essential for interacting with human papillomavirus type 31 (HPV31) E2 protein, such as the PDID and BID. Interestingly, BRD4S inhibits E2 activities upon interaction, while BRD4L activates E2-mediated viral gene transcription. Breast cancer studies have also revealed opposing functions of BRD4 isoforms. The engrailed homeobox 1 (EN1)–BRD4S axis has been identified as oncogenic, while BRD4L exhibits tumor-suppressive properties, highlighting the context-dependent roles of these long and short isoforms. Given that the functionality of BET proteins primarily relies on their C-terminal regions and associated interactomes, it becomes crucial to investigate and comprehend the functional distinctions between BRD4L and BRD4S in various disease contexts [15].

## 3. BRD4 in Aging-Related Vascular Pathologies and Diseases

Although BRD4 was initially recognized for its involvement in cancer progression, its significance has been found in many other diseases, such as heart disease and infectious diseases. Recent studies have also revealed that the aberrant expression and impaired function of BRD4 were highly associated with aging vascular pathology, affecting multiple key biological processes in the vascular cells and tissues, providing new insights into the understanding of vascular pathophysiology and pathogenesis of vascular diseases. Despite the limited information and insufficient knowledge, the available research results indicate that BRD4 may play a critical role in the pathogenesis of many aging-associated vascular diseases and could be a potential therapeutic target; thus, we summarized the most recent studies of BRD4 in the relevant studies as below.

### 3.1. Atherosclerosis

Atherosclerosis is a chronic vascular disease that is characterized by lipid buildup in the intima, thickening of the arterial wall, and constriction of the vascular cavity, all of which can result in serious cardiovascular complications [51]. Dysregulated lipid and fat metabolism are the key risk factors for arteriosclerosis. It has been shown that BRD4 is required for the activation of peroxisome proliferator-activated receptor gamma 2 (PPARγ2) and CCATT-enhancer-binding protein alpha (C/EBPα) gene transcription during adipocyte differentiation [52], affecting the lipid metabolism in the vascular cells, contribute to the pathogenesis of atherosclerosis.

In addition, it has been shown that BRD4 participates in macrophage senescence and generates senescence-associated secretory phenotype (SASP), which furthers the development of atherosclerosis. In a study that gathered a variety of macrophages from several sources, increasing BRD4 expression induced cellular senescence by encouraging the formation of SASP and the advancement of atherosclerosis-like conditions in the models with lipopolysaccharide (LPS) treatment. Following infection, the cells expressed more NF-ĸB-dependent BRD4, which encouraged the development of inflammatory factors and ultimately caused macrophage senescence via an autocrine route. Reciprocally, treatments with siBRD4 and BRD4 inhibitors, such as JQ-1 and selective inhibitor of bromodomain extra-terminal domain 762 (I-BET762), shield the cells from senescence brought on by LPS. Meanwhile, the inhibition of BRD4 by siBRD4 or inhibitors can stop the spread of senescence from older to younger cells, preventing paracrine SASP production. Further research needs to be done on novel BRD4 pathways that contribute to autocrine or paracrine senescence in vivo [53]. This data indicates that BRD4 is a promising pharmacological target for atherosclerosis and disorders associated with aging.

### 3.2. Aortic Aneurism

Abdominal aortic aneurysm (AAA) is a threatening vascular disease due to acute aortic rupture in 90% of patients [54]. A recent study identified that miR-124a was an important regulator in AAA progression, and BRD4 was determined to be the downstream target of miR-124a in AAA cells. It was also found that miR-124a was significantly downregulated in the whole blood of AAA patients and AAA cell models [49]. Overexpressing miR-124a could effectively inhibit the proliferation and migration and promote the apoptosis of the AAA cells, and upregulation of BRD4 could reverse the miR-124a effects on AAA cell phenotype. Additionally, it has been shown that miR-124A has the potential of wingless-related integration site/ beta- catenin (Wnt/β-catenin) and P53 activity regulation via targeting BRD4. Moreover, given data on miR-124a suggests it can regulate the activities of Wnt/β-catenin and P53 to repress AAA progression by targeting BRD4 [49].

### 3.3. Vascular Neointima Formation

Neointima formation is an important pathological manifestation of vascular remodeling related to various cardiovascular diseases and is also a major cause of failure in revascularization surgery. Enhanced migration and proliferation of vascular smooth muscle cells (VSMCs) is the main etiology of the neointima formation [55]. Studies have shown that in rat balloon angioplasty models, BRD4 was significantly increased in neointima formation, and blocking BRD4 with JQ-1 inhibited intimal hyperplasia (IH), likely by diminishing proliferation and migration of VSMCs. Treatment with JQ-1 also prevented cytokine-induced apoptosis and impairment of human vascular endothelial cells (ECs) [56]. Human recombinant platelet-derived growth factor subunit B (PDGF-BB) increases BRD4 protein levels, suggesting a role in the phenotypic transition of rat primary aortic VSMCs stimulated by PDGF-BB. Indeed, JQ-1 treatment induced a pronounced inhibitory effect on the proliferation and migration of rat VSMCs and human primary aortic VSMCs [56]. Evidence has been shown that BRD4 modification might diminish neointima development in vein grafts in vivo [57,58].

### 3.4. Pulmonary Arterial Hypertension (PAH)

PAH is a progressive disorder characterized by high blood pressure in the arteries of the lungs (pulmonary arteries), which are the blood vessels that carry blood from the right ventricle through the lungs, leading to right heart failure and premature death. Accumulating evidence suggests that BRD4 is implicated in the pathogenesis of PAH and targeting BRD4 represents a novel therapeutic approach for the treatment of PAH. The relevant studies regarding the therapeutic effect of BET inhibition on PAH vascular remodeling have been well summarized in a recent review [59].

It has been shown that BRD4 is markedly increased in pulmonary vascular lesions and hypertrophic right ventricular (RV) tissue of patients with PAH and Sugen5416/hypoxia (SuHx)-exposed rats [60,61]). Nebulization of the pan-BET inhibitor JQ-1 inhibits PAH and RV hypertrophy in SuHx rats [60]. These observations were further confirmed by another multiple-center preclinical study by three independent groups [62]. BRD4 was further found to be upregulated in pulmonary microvascular endothelial cells (MVECs) and SMCs isolated from idiopathic PAH patients compared to healthy control persons. While BRD4 localized mostly in the endothelium in control individuals, it was concentrated in neointimal fibrotic lesions, lining the plexus channels inside plexiform lesions, and the media of bigger pulmonary arteries with medial hypertrophy in idiopathic PAH.

BRD4 inhibition by JQ-1 and siRNA has been found to reverse vascular remodeling and enhance pulmonary hemodynamics in Sugen hypoxia-pulmonary arterial hypertension rats (SH-PAH), with decreased PAH SMC apoptosis resistance and proliferation. In this study, BRD4 and forkhead box M1 (FoxM1) were suppressed with siRNA, which both reduced polo-like kinase 1 (PLK1) protein expression; and it further showed that BRD4 controls PLK1 through FoxM1 in PAH-SMCs. In PAH-SMCs, inhibition of BRD4 by

RVX208, the only BET inhibitor in phase 3 clinical trials, also resulted in a reduction in FoxM1 and PLK1 protein levels and transcriptional levels [63].

Additionally, studies indicated that BRD4 is a known trigger of Runt-related transcription factor 2 (*RUNX2*), and in some cancer cells, *RUNX2* is a direct target of BRD4 inhibition by JQ-1 [64,65,66]. Interestingly, *RUNX2* is found to be upregulated in the lungs, distal pulmonary arteries, and primary cultured human pulmonary ASMCs isolated from PAH patients and plays a pivotal role in the pathogenesis or the development of proliferative and calcified pulmonary artery lesions [63]. This data together implies that BRD4 may affect the initiation and development of PAH by regulating *RUNX2*.

### 3.5. Essential/Systemic Hypertension

Hypertension is one of the most common vascular diseases with complicated causes, resulting in various heart diseases and strokes. Several lines of evidence indicate the potential association between BRD4 and systemic hypertension in human and animal models. It has been reported that blood BRD4 in essential hypertension (EH) patients is higher than in the healthy control [67] and was positively correlated to systolic and diastolic blood pressure of enrolled subjects including patients with EH and healthy controls. Inhibiting BRD4 could reduce oxidative stress and inflammatory response, alleviate endothelial cell damage, ameliorate aortic injury, and lower blood pressure, indicating that BRD4 inhibition could be a potential target for the clinical treatment of EH patients.

In spontaneously hypertensive rats (SHRs), an increased BRD4 is also reported by several groups [67]. BRD4 inhibiter JQ-1 treatment significantly decreased blood pressure in SHR rats, without significant changes in food intake and body weight, suggesting a potential role for BRD4 in the development of hypertension [67]. Mechanistically, JQ-1-treated rats had remarkably lower levels of plasma Angiotensin II (AngII), endothelin-1 (ET-1), nitric oxide (NO), and nitric oxide synthase (NOS). JQ-1 remarkably enhanced plasma superoxide dismutase (SOD) activity and reduced the content of malondialdehyde (MDA), *IL-6,* and tumor necrosis factor-alpha (TNF-α) in SHR rats. These results indicated that JQ-1 may lower blood pressure by reducing oxidative stress and inflammatory responses in SHR rats [67].

In addition, AngII-induced hypertension is known to be one of the most common models used in animal studies, which has been linked to many other cardiovascular diseases [68]. BRD4 was found to be increased in mice treated with AngII. Treatment with BRD4 inhibitor JQ-1 ameliorates AngII-induced hypertension, medial hypertrophy, and inflammation in vivo in mice [69], further underscoring the promise of such epigenetic modifiers for the treatment of human hypertension. Mechanistically, it was found that AngII stimulation increases the recruitment of TFs in VSMCs, such as activator protein 1 (*AP1*), erythroblast transformation-specific *ETS,* signal transducer and activator of transcription 1 (*STAT1*), and *NF-κB*, at their respective binding sites on the enhancers/SEs and subsequent downstream signaling [69]. Subsequently, BRD4 is recruited, resulting in the addition of histone 3 lysine 27 acetylation (H3K27ac) enrichment and SE formation, thereby enhancing the transcription of nearby genes in VSMCs [69]. Several studies have revealed that BRD4 can function as a scaffold for transcription factors in the promoters and super-enhancers (SEs) [70]. Moreover, AngII stimulation also recruits other TFs, such as caudal-type homeobox 2 (*CDX2*), *FOXL1*, and *LIN54*, leading to a loss of H3K27ac enrichment on the enhancers to downregulate the associated genes, which could be disrupted by BRD4 inhibitor JQ-1 [71,72]. This data together provides a new perspective on the epigenetic mechanisms underlying AngII’s actions and VSMC dysfunction, including previously unrecognized roles of enhancers and SEs in VSMCs, and their potential connections to CVDs. This data could also be exploited for drug discovery. Despite these findings, the studies on BRD4 in systemic hypertension are much fewer so far.

## 4. Conclusions

Increasing evidence indicates that BRD4 acts as an important epigenetic reader participating in various critical cell processes by acetylating specific histones and binding to TF and the splicing regulators, regulating gene expression in cells. Previous studies on BRD4 focused on cancer and autoimmune disease and have suggested that BRD4 may become a potential therapeutic target [22]. Until recently, BRD4 was recognized to be potentially involved in several aging-related vascular diseases, such as atherosclerosis, aortic aneurysm, vascular neointima formation, PAH, and EH, indicating a potential aging-related epigenomic mechanism underlying the pathogenesis of vascular diseases. Mechanistic studies also reveal several of the regulation of BRD4 in the vascular pathological processes, such as adipocyte differentiation, macrophage senescence, inflammation, oxidative stress, VSMC proliferation, migration, calcification and apoptosis, and EC dysfunction. Emerging evidence has also suggested that BRD4 inhibition possibly delays or even represses these pathological processes. BET inhibitors such as JQ-1, and related small molecules that interfere with SEs, are effective in animal models of atherosclerosis, AAA [73], inflammatory renal disease [73], PAH [60], and carotid intimal hyperplasia [56], highlighting the promise of such epigenetic modifiers for the treatment of human vascular diseases.

However, despite these exciting advances, there are many knowledge gaps that need further investigation. First, although more and more evidence has indicated the critical role of BRD4 in aging-related diseases, the role of BRD4 in the natural aging process is much less reported. A recent study showed that BET inhibition interferes with the association of BRD4, p300, and acetylated histone H4K16 with the Nox4 promoter in lung fibroblasts stimulated with the profibrotic cytokine, TGF-β1 in aged mice [74]. Another relative study showed that BRD4 may be involved in the transformation resistance in a premature aging disorder, and inhibits, albeit to a lower extent, the tumorigenic potential of transformed cells from healthy individuals [75]. Despite these limited studies, the regulatory mechanisms of BRD4 in “healthy aging” remain largely known. Second, at present, many BET BD inhibitors are in active clinical trials for cancers, and several have achieved or are achieving clinical success as anticancer agents [76,77]. However, no current inhibitor is selective for BRD4 due to structural similarity among the family members. Third, numerous pieces of evidence from preclinical studies or clinical trials have indicated that BRD4 may represent a promising therapeutic target for aging-related cardiovascular diseases, such as PAH, HF, and atherosclerosis, as a potent suppressor of inflammatory and fibrotic gene transcription. However, concerns were raised due to contrary reports on the biological function of BRD4 in multiple cells. For example, preclinical models have demonstrated the therapeutic efficacy of JQ-1 in limiting HF progression [31]. However, the deletion of BRD4 specifically from cardiomyocytes has been shown to have detrimental effects, resulting in spontaneous DCM. In addition, as BRD4 has a protective role in lung and breast cancer, it can promote leukemia and lymphoma. These results highlight the specific nature of BRD4’s function, which depends on the type of cancer and the tissue involved. Considering the interest in targeting BRD4 for therapy, it is crucial to carefully assess its potential effects on different cell types and tissues [75]. Furthermore, serval unanticipated side effects were noticed in clinical trials, which raised significant safety concerns. For example, JQ1 has demonstrated the ability to disrupt the BRD4-P-TEFb interaction and thereby reactivate HIV-1 transcription in latent infected human T cells. Thus, a better understanding of the biological function of BRD4 is necessary to prevent dangerous side effects in clinical trials and future applications. Therefore, an in-depth understanding of the precise mechanisms underlying BET BD inhibition and the discovery of more potent inhibitory compounds will foster the development of therapeutic strategies for vascular aging and human vascular diseases. The collected and discussed information has been summarized in Figure 2 and Table 1.

## Figures and Tables

**Figure 1 biomolecules-13-01135-f001:**
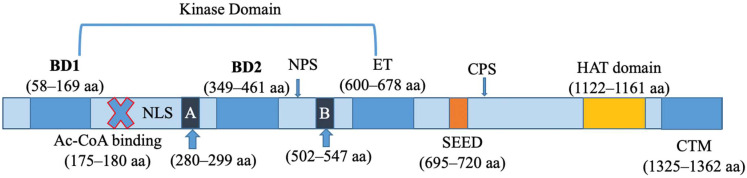
**The molecular structure of human BRD4**. BRD4 consists of serval functional domains, binding sites, and specific motifs/residues. BD1 and BD2: bromodomain 1 and 2; NLS: nuclear localization signal site; NPS: N terminal cluster of phosphorylation sites; ET: extra-terminal domain; A and B: conserved motifs; SEED: Ser (S)/Glu (E)/Asp-rich region; CPS: C-terminal cluster of phosphorylation sites; HAT: histone acetyltransferase catalytic domain; CTM: C-terminal motif.

**Figure 2 biomolecules-13-01135-f002:**
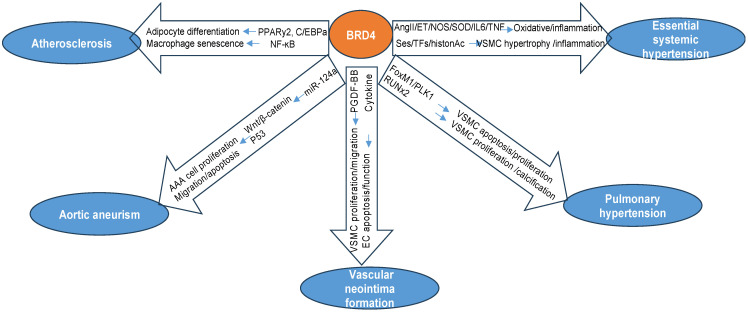
**Summary of the potential regulation of BRD4 in vascular diseases.** Current studies indicate that BRD4 has been involved in various vascular diseases through diverse mechanisms in regulating vascular function and remodeling via multiple molecular signaling.

**Table 1 biomolecules-13-01135-t001:** The models and key findings from the studies of vascular pathologies.

S/N		RESULTS	REFERENCES
1	**Ang II Infusion**	**A**. Increased BRD4 are reported by several groups in SHR rats and mice treated with *AngII*. **B**. *AngII* stimulation increased the recruitment of transcription factors at their respective binding sites. BRD4 is recruited, enhancing the transcription of nearby genes in VSMCs. **C**. Early inflammation was produced by *AngII*; by 3 days, more infiltrating interstitial monocytes/ macrophages and proinflammatory mediators were seen than in mice treated with JQ-1.	**1**. [67] Yang et al. 2018 **2**. [69,70,71]; Brown et al. 2014 Loven et al. 2013 Das et al. 2017 **3**. [72] Suarez-Alvarez et al. 2017
2	**Carotid Artery Angioplasty**	**A**. BRD4 was significantly increased in neointima formation, and blocking BRD4 with JQ-1 inhibited IH, by diminishing proliferation and migration of VSMCs. JQ-1 treatment induced a pronounced inhibitory effect on the proliferation and migration of rat and human primary aortic VSMCs.	**1**. [56] Wang et al. 2015
3	**Adipogenesis /Myogenesis cell Differentiation**	**A**. BRD4 functions as an enhancer epigenomic reader that links active enhancers to the activation of the cell identity gene during differentiation. BRD4 is required for BAT and muscle development, as evidenced muscle mass were severely reduced when BRD4 is deleted in the cervical regions of E18.5 embryos. **B**. RNA interference-based specific BET protein knockdown demonstrated that BRD4 was necessary for myogenic differentiation.	**1**. [23] Lee et al. 2017 **2**. [24] Roberts et al. 2017
4	**EndoMT Vein Graft**	**A**. BRD4 control of mediators and indicators of EndoMT, showed that BRD4 modification might diminish neointima development in vein grafts in vivo given recent evidence that EndoMT is essential for neointima formation. **B**. MicroRNA -145 encouraged the binding of myocardin to SRF. Additionally, it suppressed KLF4, which through blocking myocardin and SRF, promoted aberrant VSMC development.	**1**. [57] Zhang et al. 2019 **2**. [58] Verma et al. 2013
5	**Infection-Induced Senescent Macrophages**	**A**. By increasing BRD4 expression, the models used for LPS treatment showed to induce cellular senescence by encouraging the formation of SASP and the advancement of atherosclerosis-like conditions.	**1**. [53] Wang et al. 2020
6	**Abdominal Aortic Aneurysm (AAA) cells**	**A**. Downregulation of miR-124a was present in whole blood of patients and decreased in AAA cell models. The confirmation of the dual-luciferase reporter assay depicted that BRD4 was a downstream target of miR-124a, and the upregulation of BRD4 could potentially reverse the miR-124a effects on AAA cell phenotype.	**1**. [49] Liu et al. 2022

## Data Availability

Data sharing is not applicable. No new data were created or analyzed in this study. Data sharing is not applicable to this article.

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
