# Peer review of "Epigenetic Reader Bromodomain-Containing Protein 4 in Aging-Related Vascular Pathologies and Diseases: Molecular Basis, Functional Relevance, and Clinical Potential"

_biomolecules, 2023, doi:10.3390/biom13071135_

Round 1
Reviewer 1 Report
This review addresses a very important and hot topic regarding the new progress in the epigenomic mechanisms related to aging-related vascular disease. BDR4 is an attractive epigenetic reader that has been shown to play essential roles in regulating normal cellular functions and development of many human diseases. The summarized information is significant to advance our knowledge of this regulator and enhance our understanding of epigenetic regulation in aging-related vascular diseases. The literature was organized logically, and the manuscript is well-written. The review provides valuable information for the researchers in the field of cardiovascular research and will stimulate new studies in relative diseases.
I have the following comments that can be considered to improve the manuscript.
1. It mentioned that aging-related vascular diseases share some pathologies that could be mediated by some common genes and mechanisms. In this case, it might be interesting to look at the BRD4 in natural aging procedures. Are there any relative studies in BRD4 in natural aging vessels or other tissues? If so, it would be beneficial to include this information.
2. Some sentences/statements are not very clear. Such as: In Section 2.2: Gene transcription Regulation, second paragraph: “Regulation of BRD4 and TF interactions have potential contributions to the effects of therapeutic transcription of pharmacological BET inhibition that has been illustrated in the state of specific diseases”. It is unclear with “the effects of therapeutic transcription of pharmacological BET inhibition that has been illustrated in the state of specific diseases”. Please rephrase it or clarify it.
3. In addition, in the Same paragraph: “CDK9 (of the P-TEFb complex) has been demonstrated to be phosphorylated by BRD4, which controls its enzymatic activity,” It is unclear what the P-TEFb complex is. Please descript it or clarify it.
4. Some abbreviations did not have their full terms in the first place of their use, such as KO, CDK9, et al. Please check through the manuscript and make corrections accordingly.
minor editing
Reviewer 2 Report
Aging is a critical risk factor for many cardiovascular diseases and cancers. A better understanding of the underlying mechanisms and involved regulatory singling is significant and necessary for exploring therapeutic targets. Emerging evidence indicates that epigenetic regulation plays an important role in aging-related pathologies in various cells and tissues. This review summarized the new progress in the studies on BRD4, a key epigenetic reader that was initially recognized in cancers and autoimmune diseases and recently attracted a lot of attention in cardiovascular diseases, advancing the new knowledge of the biological function of this epigenetic regulator in vascular tissue and cells. This review also discussed the relevance of BRD4 in aging-related vascular diseases and the therapeutical potential of its inhibition, which will open new research directions in the relative fields. The manuscript is well-organized and informative. The collected information will also be beneficial to the research in other aging-related studies. I only have following minor issues/suggestions:
1. For the clinical potential: Is there any application in the current clinical therapy or clinical trials in any diseases using BRD4 inhibitors? It would be beneficial to the readers to include the relative information. It would also be helpful to add some discussions about the potential or the prospection of the prevention or treatment of cardiovascular diseases by targeting BRD4.
2. The alternative splicing of BRD4 is very interesting and could be extended to include more detail regarding the distinctive functions between these two isoforms and their association with the diseases.
3. Please double-check all the abbreviations to ensure they are fully defined in the first place but not double-defined in the text.
Please double-check all the abbreviations to ensure they are fully defined in the first place but not double-defined in the text.
